# ZePo: Zero-Shot Portrait Stylization with Faster Sampling

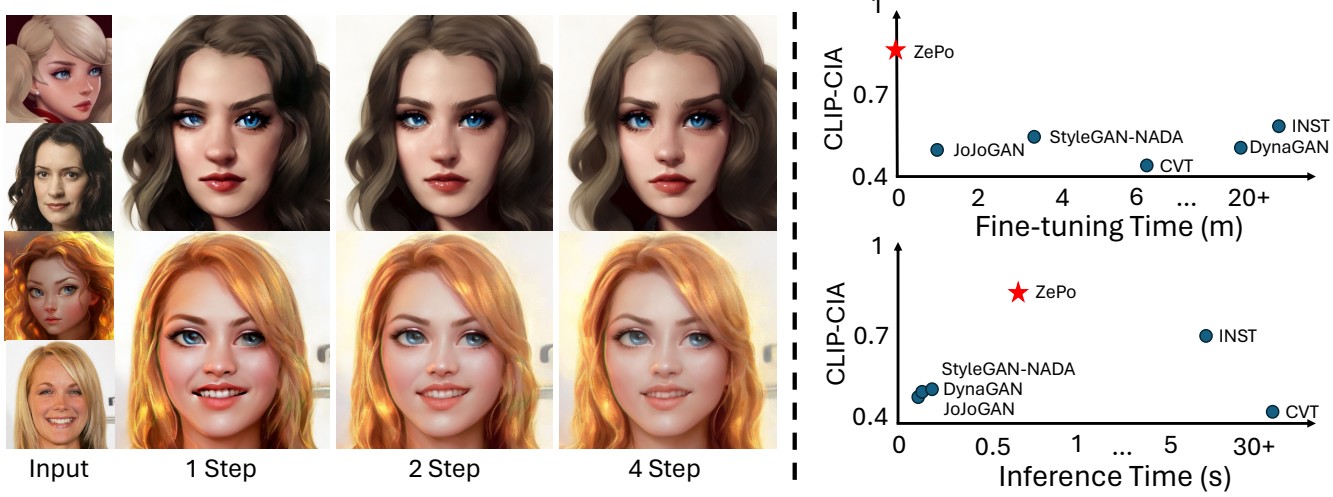

**Figure 1: The proposed zero-shot portrait stylization framework ZePo can directly synthesize stylized facial images with very few sampling steps (including 1, 2, and 4 steps) (left), where the images synthesized in 4 steps have the best overall quality as measured by the CLIP-CIA metric. Moreover, our method does not require model fine-tuning, and with 4-step sampling, the inference time is only about 0.6 seconds (right).**

## ABSTRACT

Diffusion-based text-to-image generation models have significantly advanced the field of art content synthesis. However, current portrait stylization methods generally require either model fine-tuning based on examples or the employment of DDIM Inversion to revert images to noise space, both of which substantially decelerate the image generation process. To overcome these limitations, this paper presents an inversion-free portrait stylization framework based on diffusion models that accomplishes content and style feature fusion in merely four sampling steps. We observed that Latent Consistency Models employing consistency distillation can effectively extract representative Consistency Features from noisy images. To blend the Consistency Features extracted from both content and style images, we introduce a Style Enhancement Attention Control technique that meticulously merges content and style features within the attention space of the target image. Moreover, we propose a feature merging strategy to amalgamate redundant features in Consistency Features, thereby reducing the computational load

of attention control. Extensive experiments have validated the effectiveness of our proposed framework in enhancing stylization efficiency and fidelity.

## CCS CONCEPTS

• **Computing methodologies → Image manipulation**.

## KEYWORDS

Portrait Stylization, Diffusion Model, Zero-Shot

## 1 INTRODUCTION

Portrait stylization involves the transfer of an art style from a reference portrait to a standard facial photograph. Traditional methods for portrait stylization [7, 9, 29, 51, 67] typically involve fine-tuning a pre-trained generative model [17, 46], such as StyleGAN [23] or Stable Diffusion [46], using various reference art portraits. However, these approaches necessitate considerable time for model fine-tuning and additional storage space to accommodate the models that have been fine-tuned for each distinct style image.

To overcome the limitations mentioned above, recent studies have investigated a tuning-free method [10, 11, 31] that leverages self-attention features from both content and reference images during the generation process through Attention Control [5], enabling zero-shot portrait stylization. Despite this advancement, the method struggles with slow image generation speeds. On the one hand, the diffusion model requires an extensive sampling process to iteratively denoise Gaussian noise. On the other hand, to accurately reconstruct the content and reference images, this method often

*ACM MM, 2024, Melbourne, Australia*

© 2024 Copyright held by the owner/author(s). Publication rights licensed to ACM.
ACM ISBN 978-x-xxxx-xxxx-x/YY/MM
https://doi.org/10.1145/nnnnnnn.nnnnnnn

depends on the protracted DDIM Inversion [52] process, which is necessary to obtain a sequence of intermediate anchors for image reconstruction. Additionally, the manually customized Attention Control [5] operation exacerbates the situation by involving excessive computations of the redundant self-attention mechanism, further impeding the image generation speed.

In this work, we introduce **ZePo**, a **Ze**ro-shot **Po**rtrait Stylization framework, to address the aforementioned challenges. Regarding the issue of slow sampling speeds, one intuitive solution is to employ high-order numerical ODE solvers [2, 34, 70] to decrease the number of sampling steps required for image generation. However, these methods, which leverage high-order ODE approximations, necessitate multiple network function evaluations (NFEs) and achieve only a marginal reduction in actual sampling time. Moreover, these ODE solvers do not integrate well with the intermediate anchors established by DDIM Inversion, which restricts their applicability for this particular method. Therefore, rather than relying on high-order ODE samplers, we propose the use of accelerated distillation of pre-trained models, specifically Latent Consistency Models (LCMs) [35], to expedite the image synthesis process. Additionally, to obviate the need for DDIM Inversion, our findings indicate that LCMs can directly extract representative consistency features from noised images. Building on this capability, we suggest a method to directly extract consistency features from noisy reference and content images. These features are then seamlessly incorporated during the generation process of the target image, resulting in a more efficient and streamlined stylization approach.

To address the issue of speed reduction due to redundant computations in conventional Attention Control methods, we introduce the Style Enhancement Attention Control (SEAC). SEAC begins by integrating the redundant consistency features from both the source and reference images. Subsequently, it concatenates these merged features and maps them as key and value features within the self-attention space. To modulate the degree of image stylization, the key features of the reference image are multiplied by a Style Enhancement coefficient. Consequently, the attention map, calculated using the query features from the target image and the merged key features, can adaptively select the value features from both the content and reference images. This method not only increases the computational speed of Attention Control but also mitigates the issue of query confusion, enhancing the precision and efficiency of the stylization process.

Ultimately, as illustrated in Figure 1 (left), our method demonstrates the capability to synthesize stylized portraits using no more than four sampling steps, significantly enhancing both the speed and practicality of portrait stylization using diffusion models. Through extensive experimentation, we have demonstrated the advantages of our ZePo framework in rapid stylized portrait synthesis. As illustrated in Figure 1 (right), ZePo does not require additional fine-tuning time, and it achieves the optimal CLIP-CIA score while reducing the inference time to just 0.6 seconds using a 4-step sampling process.

To summarize, we make the following key contributions:

(i) We introduce ZePo, a new inversion-free portrait stylization framework that requires as few as one sampling step to synthesize high-quality stylized portraits.

(ii) We propose a novel attention control mechanism, termed Style Enhancement Attention Control, which leverages redundant feature fusion to enhance the speed of self-attention computations and can adaptively select value features from source and reference images.

(iii) We demonstrate from both quantitative and qualitative perspectives that our method surpasses existing state-of-the-art baselines, achieving a significantly better balance between preserving source content information and enhancing image stylization.

## 2 RELATED WORKS

### 2.1 Few-Shot Face Stylization

Early methods [20, 26, 33, 57, 72, 73] of face stylization often required sampling a large volume of image data to train image-to-image translation models, consuming substantial training resources. To mitigate the high cost of training and capitalize on the prior knowledge embedded in pre-trained models, the few-shot face stylization approach has gained considerable attention. This method involves fine-tuning a pre-trained StyleGAN model [22–24] with a limited number of target image samples, a technique commonly known as GAN-adaptation [39, 45, 58, 59, 61, 69, 71]. Toonify [42] was among the pioneers in experimenting with face stylization using GAN-adaptation. They initially fine-tune a StyleGAN model using a limited set of cartoon samples and subsequently interpolated the weights of the fine-tuned model with those of the original model to generate cartoon-styled faces. [30, 40] introduced additional regularization terms in the latent space to mitigate the tendency of overfitting when fine-tuning pre-trained models with a small number of samples. AgileGAN [51] introduced an inversion-consistent transfer learning framework that effectively reduces the variance in the inversion distribution. [60] developed a method that introduces an intermediate domain between the source and animation domains to bridge the gap between the two. DualStyle-GAN [63] adds an additional style path to a pre-trained StyleGAN, enabling efficient modeling and adjustment of both intrinsic and extrinsic styles. However, it still requires hundreds of images for fine-tuning, which limits its applicability in scenarios with very few examples. JoJoGAN [9] advances this by proposing a one-shot face stylization method that utilizes a reference image to generate a style mixed paired dataset. The model is then fine-tuned using this dataset with pixel loss, enhancing its utility in limited-sample environments. [68] proposed a novel one-shot adaptation method for face stylization, which divides the process into style transformation and identity transformation. By effectively separating identity from style, their approach results in more natural and coherent transformation outcomes. StyleDomain [1] introduced a parameter-efficient method that adapts pre-trained models to new domains by modifying style vectors within the Style Space, enhancing adaptability with minimal resource usage. [71] utilize a single real-style paired reference to provide style direction in the DINO-ViT [6] feature space, enabling precise fine-tuning of the generative model.

### 2.2 Diffusion-Based Style Transfer

Diffusion models [12, 18, 50, 52, 54] have increasingly become prominent in the field of generative models in recent years, particularly with the advent of pre-trained text-to-image (T2I) models

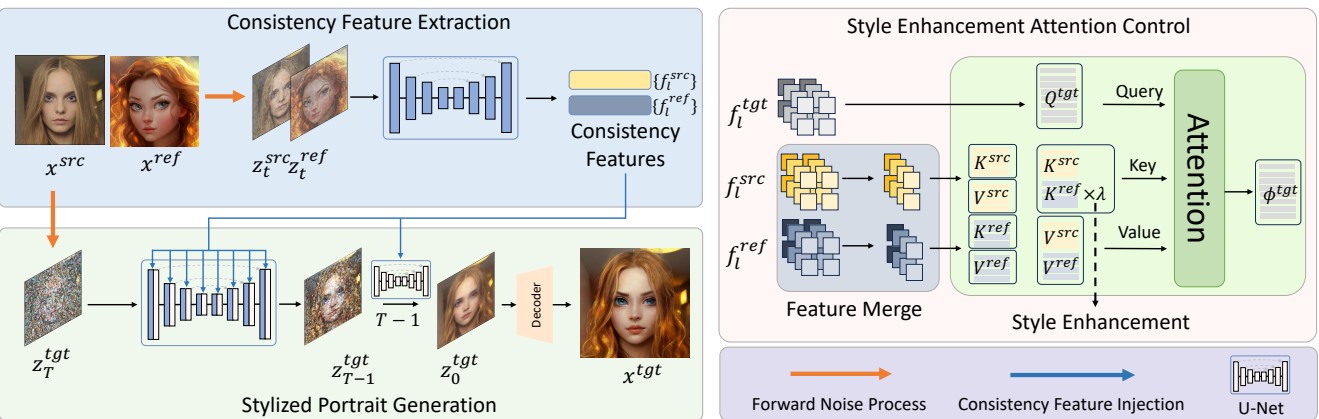

**Figure 2: The overall framework of ZePo. The framework is divided into two stages. The first stage involves the extraction of consistency features, where multi-scale consistent features are extracted from the reference and source images with slight noise added. The second stage is the stylized image synthesis phase, where the source image, added with a moderate level of noise, is used as the input. In this phase, the Style Enhancement Attention Control module within the U-Net fuses the consistency features from both the reference and source images to synthesize a stylized portrait.**

[41, 46, 49]. These models have not only popularized AI-generated art but also spurred extensive research into style transfer methods leveraging diffusion models. Several methods are inspired by classifier guidance [12], utilizing precisely formulated energy functions to deliver gradient information for guiding image generation [38, 64]. This allows unconditional diffusion models to produce images conditioned on specific content and style. For instance, [28] advocates for employing the style loss derived from a pre-trained DINO-ViT [6] to guide the generation of stylized images. Similarly, [62] utilizes style loss generated by CLIP [44] for guiding stylized image generation. Other techniques focus on personalizing diffusion models with specific styles by fine-tuning pre-trained T2I models using approaches like LoRA [19], Textual Inversion [14], or Dreambooth [48]. Subsequently, DDIM Inversion [43, 52] is used to derive a noise representation of content images, which the fine-tuned model denoises in the noise space, thereby generating stylized images [7, 25, 67]. Particularly, [7] combines this with null-text inversion [36] to achieve more precise content reconstruction, though this method tends to slow down the image synthesis process. Distinct from methods that merely fine-tune T2I models, some strategies [8] involve fine-tuning a pre-trained Diffusion Autoencoder [43] using optimized semantic latent codes to meticulously control both content and style. Furthermore, techniques such as ControlNet [65] and T2I-Adapter [37] train additional style adapters using extensive datasets of style images. These adapters are designed to adjust the style of images produced by pre-trained T2I models, offering a tailored approach to style management in image generation. Recent studies have begun to explore zero-shot stylization methods that utilize pre-trained T2I diffusion models [10, 11, 31]. These methods integrate content and style features within the attention space through meticulously engineered attention control modules [5]. However, they depend on DDIM Inversion to derive noise representations of content and style images, which consequently slows down the image synthesis process.

## 3 PRELIMINARIES

### 3.1 Latent Diffusion Models

Latent Diffusion Models (LDMs) [46] employ a diffusion model within the latent space of a pre-trained Variational Autoencoder (VAE) [13]. The encoder $\mathcal{E}$ encodes images into latent codes $z_0 = \mathcal{E}(x)$, while the decoder $\mathcal{D}$ reconstructs images $x = \mathcal{D}(z_0)$ from these codes.

The forward process of diffusion models operates as a Markov chain, incrementally introducing noise into the initial latent code $z_0$. Due to the additive nature of Gaussian noise, this process is generally modeled as a single-step addition of noise, directly yielding the noisy latent code $z_t$ at any given step $t$:

$$\mathbf{z_t} = \sqrt{\alpha_t}\mathbf{z_0} + \sqrt{1-\alpha_t}\boldsymbol{\epsilon}, \quad \boldsymbol{\epsilon} \sim \mathcal{N}(\mathbf{0}, \mathbf{I}), \quad (1)$$

where $\alpha_t$ is a predefined diffusion schedule. The reverse process of diffusion models constitutes an approximate Markov chain, where progressively removing noise in $z_T$ through the reverse process, ultimately restoring the noise-free latent code $z_0$ after $T$ iterative steps:

$$\mathbf{z_{t-1}} = \frac{1}{\sqrt{1-\beta_t}}(\mathbf{z_t} - \frac{\beta_t}{\sqrt{1-\alpha_t}}\boldsymbol{\epsilon}_\theta(\mathbf{z_t}, t)) + \sigma_t\boldsymbol{\epsilon}, \quad (2)$$

where $\boldsymbol{\epsilon}_\theta$ is a time-conditioned U-Net [47], tasked with predicting the noise component in $z_t$ at each step $t$. The parameters $\theta$ within $\boldsymbol{\epsilon}_\theta$ are fine-tuned by minimizing a noise prediction loss:

$$L(\theta) = \mathbb{E}_{t,z_0,\epsilon}\left[\|\boldsymbol{\epsilon} - \boldsymbol{\epsilon}_\theta(z_t, t)\|^2\right], \quad (3)$$

where $\boldsymbol{\epsilon}$ denotes the noise introduced during the forward process as described in Eq. 1.

### 3.2 Latent Consistency Models

Latent Consistency Models (LCMs) [35], are a specialized form of Consistency Models (CMs) [53] that significantly accelerate the generation speed of LDMs. In LCMs, the consistency function $f(z_t, t)$ ensures that each anchor point $z_t$ in the sampling trajectory can

be accurately mapped back to the initial latent code $z_0$, thereby ensuring self-consistency within the model. The consistency function is defined as follows:

$$f(x, t) = c_{\text{skip}}(t)x + c_{\text{out}}(t)F(x, t), \qquad (4)$$

where $c_{\text{skip}}(t)$ and $c_{\text{out}}(t)$ are differentiable functions designed to ensure the differentiability of $f(x, t)$ with conditions $c_{\text{skip}}(0) = 1$ and $c_{\text{out}}(0) = 0$. The efficacy of $f(x, t)$ is measured through the following optimization objective:

$$\min_{\theta, \theta^-; \phi} \mathbb{E}_{z_0, t} \left[ d \left( f_\theta(z_{t+1}, t+1), f_{\theta^-}(\hat{z}_t^\phi, t) \right) \right], \qquad (5)$$

where $f_\theta$ denotes a consistency function parameterized by a trainable neural network, and $f_{\theta^-}$ is updated at a slow decay rate $u$ to adjust parameters within $f_\theta$. The variable $\hat{z}_t^\phi$ represents a one-step estimate of $z_t$ obtained through the sampler $\phi$ from $z_{t+1}$.

## 4 METHOD

In this section, we introduce **ZePo**, a zero-shot framework for portrait stylization that operates within four sampling steps. Our framework leverages Latent Consistency Models (LCMs), a variant of Stable Diffusion that distilled with the consistent objective (Eq. 5). ZePo capitalizes on the observation that LCMs not only significantly reduce the number of sampling time steps required for generating images but also efficiently extract representative features from noisy images, which we term Consistency Features. Utilizing the Consistency Features extracted from both source and reference images, we seamlessly integrate these features into the image generation process through our proposed Style Enhancement Attention Control module. This integration allows for subtle yet effective stylization adjustments. Ultimately, with just four sampling steps, our framework is capable of synthesizing high-quality stylized portraits that faithfully capture the style of the reference image. The overall architecture of our framework is depicted in Figure 2.

### 4.1 Consistency Features

The primary purpose of employing DDIM Inversion [52] is to derive a series of anchor points $\{z_t\}$ that facilitate the reconstruction of the original image $z_0$, where each anchor $z_t$ is capable of recovering $z_0$ with better accuracy. As illustrated in Fig. 3 (a) (b), utilizing the noisy latent $z_t$ derived from the forward process in Eq. 1, tends to yield a predicted $z_0$ that is blurry and lacks high-frequency details. In contrast, the noisy latent $z_t$ post DDIM Inversion can estimate $z_0$ with enhanced accuracy. The optimization objective (Eq. 5) of LCMs is aims to minimize the disparity between the outputs of consistency function in adjacent samples, which corresponds to the distance between one-step predictions of the model for $z_0$. It is observed that this objective endows LCMs with superior one-step predictive capabilities for $z_0$. As depicted in Figure 3 (c), the noise level during forward process is relatively low, particularly for time steps $t \leq 300$, the estimated $\hat{z}_0$ by LCM exhibits clearer and more consistent details compared to the original $z_0$. This suggests that LCMs can effectively extract representative features from a noisy image, which is referred to Consistency Features. Inspired by this capability, we propose leveraging the Consistency Features extracted from both

source and reference images for portrait stylization. This method effectively replaces the time-consuming DDIM Inversion process, offering a more efficient pathway to achieving high-quality portraits stylization.

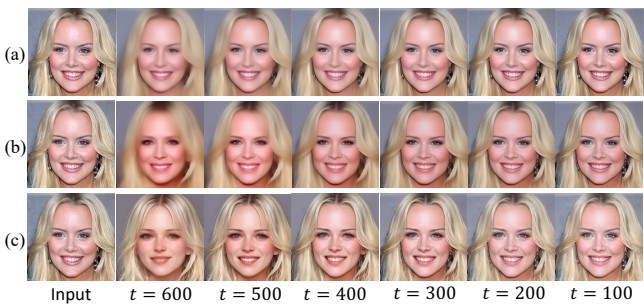

**Figure 3: The results of one-step denoising with different noise levels (time-step), different noise addition methods (DDIM Inversion and Forward Process), and different models (SD and LCM) are examined. (a) DDIM Inversion + SD. (b) Forward Process + SD. (c) Forward Process + LCM.**

Given a source image $I^{src}$ and a reference image $I^{ref}$, initially, a pre-trained VAE encoder $\mathcal{E}$ encodes them into latent codes $z^{src}$ and $z^{ref}$, respectively. Subsequently, a forward process (Eq. 1) is applied to introduce noise to these latent codes in a single step, defined as follows:

$$z_t^{src} = \sqrt{\alpha_t} z_0^{src} + \sqrt{1 - \alpha_t} \epsilon,$$
$$z_t^{ref} = \sqrt{\alpha_t} z_0^{ref} + \sqrt{1 - \alpha_t} \epsilon,$$

where $t$ represents a smaller time-step, and $\epsilon \sim \mathcal{N}(0, I)$. Finally, the noisy latent codes $z_t^{src}$ and $z_t^{ref}$ are inputted into the noise prediction network $\epsilon_\theta$ of the LCMs, from which the consistency features $\{f_l^{src}\}$ and $\{f_l^{ref}\}$ of the source and reference images at each transformer layer $l$ of $\epsilon_\theta$ are extracted. This process is formalized as:

$$(\{f_l^{src}\}, \{f_l^{ref}\}) = \epsilon_\theta((z_t^{src}, z_t^{ref}), t, c),$$

where $c$ denotes the textual condition.

In contrast to previous approaches [5, 55] which necessitate feature injection to align with the current generation time step, our proposed consistency features exhibit flexibility in this regard. They are not bound by the requirement to match the current generation time step. Thus, the extracted consistency features can seamlessly integrate into the generation process at any time step, ensuring their consistent contribution throughout various stages of the generation process. We demonstrate the impact of feature extraction at different time steps on the generated results in Figure 7.

### 4.2 Style Enhancement Attention Control

**Attention Control** Attention Control (AC) replaces the key and value features in the target image generation branch with those derived from the source image reconstruction branch. Leveraging the self-attention mechanism, AC adaptively aggregates features from the reference image, thereby preserving both semantic and

structural information from the source image. However, the incorporation of AC significantly impacts the speed of image generation in existing methods. We conducted a comparative analysis, measuring the time required for image generation with and without AC under identical step settings. As presented in Table 1, which indicates approximately a 30% increase in time consumption when AC is employed.

|  | T=50 | T=25 | T=10 |
|---|---|---|---|
| w/o AC | 06.55 | 03.00 | 01.22 |
| W AC | 08.61 | 04.17 | 01.64 |

**Table 1: The image generation speeds with and without Attention Control (AC) at different time steps.**

**Feature Merge** In Vision Transformers, there exists redundancy in tokens, and pruning these redundant tokens during inference can lead to a model with faster inference speed [3]. Similar techniques have been investigated within the diffusion model framework, which extensively employs self-attention modules. Merging redundant features in diffusion models has been shown to significantly enhance the speed of image generation without compromising the quality of the generated images [4]. Building upon this observation, we propose leveraging the token merge technique to merge redundant feature sequences before to attention control, thereby reducing the length of features from $N$ to $N/2$ or less. In contrast to the approach outlined in [4], which necessitates the unmerging of merged token sequences to restore the original length of token sequences, our method exclusively merges the consistency features inputted into attention control. This targeted merging strategy helps circumvent errors that may arise during the un-merging process.

**Style Enhancement Attention Control** We denote the merged consistency features at layer $l$ as $(f_l^{\hat{src}}, f_l^{\hat{ref}})$. Upon entering the Attention Control mechanism, these merged features are individually mapped to the key $(K^{src}, K^{ref})$ and value $(V^{src}, V^{ref})$ features within the self-attention module. In contrast to the conventional AC methods that directly replace key and value features, we introduce a Style Enhancement Attention Control (SEAC) mechanism. Specifically, we concatenate $(K^{src}, K^{ref})$ and $(V^{src}, V^{ref})$ from the source and reference images into a unified set of key and value features. Moreover, we enhance $K^{ref}$ by multiplying it with a Style Enhancement coefficient $\lambda$, yielding a new set of key and value features as follows:

$$K^{sr} = \text{Concat}(K^{src}, \lambda \cdot K^{ref}) \in \mathbb{R}^{B,N,D}$$

$$V^{sr} = \text{Concat}(V^{src}, V^{ref}) \in \mathbb{R}^{B,N,D}.$$

Subsequently, the key feature $K^{sr}$ and the query feature $Q^{tgt} \in \mathbb{R}^{B,N,D}$ from the target image are utilized to compute a self-attention map $A$ given by:

$$A = \text{SoftMax}\left(\frac{Q^{tgt} \cdot K^{srT}}{\sqrt{d}}\right) \in \mathbb{R}^{B,N,N},$$

where $d$ represents the dimensionality of the query and key features. Finally, the self-attention map $A$ is applied to the value feature $V^{sr}$

to derive the final output $\phi^{tgt}$ as follows:

$$\phi^{tgt} = A \cdot V^{sr}.$$

Hence, SEAC can effectively assess the similarity between the query features $Q^{tgt}$ and the combined key features $(K^{src}, K^{ref})$, enabling the adaptive aggregation of value features from $(V^{src}, V^{ref})$. Additionally, the lengths of the query, key, and value features utilized in the attention computation are all $N$, ensuring consistency in the computational cost of attention control compared to the original self-attention mechanism. The comprehensive pipeline of the Style Enhancement Attention Control is illustrated in Fig. 2 (right).

Building upon the aforementioned consistency feature and Style Enhancement Attention Control, we introduce a rapid portrait stylization framework **ZePo**. With this framework, we achieve the synthesis of stylized faces within four sample steps. The complete algorithm is detailed in Algorithm 1.

---

**Algorithm 1** Zero-shot Portrait Stylization

---

**Require:**
  Distilled Diffusion Model $\epsilon_\theta$, Encoder $\mathcal{E}$, Decoder $\mathcal{D}$;
  Prompt condition $c$, Guidance scale $s$, Sample steps $T$;
  Reference image $I^{ref}$, Source image $I^{src}$, Consistency feature step $\tau$,;
1: $z_0^{ref}, z_0^{src} \longleftarrow \mathcal{E}(I^{ref}, I^{src})$;
2: Sample noise $\epsilon \longleftarrow \mathcal{N}(0, \mathbf{I})$;
3: $(z_\tau^{ref}, z_\tau^{src}) \longleftarrow \text{Forward}((z_0^{ref}, z_0^{src}), \tau, \epsilon)$;
4: $(\{f_l^{ref}\}, \{f_l^{src}\}) \leftarrow \epsilon_\theta\left((z_\tau^{ref}, z_\tau^{src}), \tau, c, s\right)$;
5: $z_0^{tgt} \leftarrow z_0^{src}$
6: $t = T$
7: **repeat**
8:   $t = t - 1$
9:   Sample noise $\epsilon \longleftarrow \mathcal{N}(0, \mathbf{I})$;
10:   $z_t^{tgt} \longleftarrow \text{Forward}(z_0^{src}, t, \epsilon)$;
11:   $\epsilon^{tgt} \leftarrow \epsilon_\theta\left(z_t^{tgt}, t, c, s, (\{f_l^{ref}\}, \{f_l^{src}\})\right)$;
12:   $z_0^{tgt} \leftarrow \text{Prediction}(z_t^{tgt}, t, \epsilon^{tgt})$;
13: **until** $t < 0$
14: **return** $I^{tgt} \longleftarrow \mathcal{D}(z_0^{tgt})$

---

## 5 EXPERIMENTS

**Implementation Details.** Our experiments utilize Latent Consistency Models (LCMs), a variant of the acceleration-distilled Stable Diffusion, employing the LCM sampler. The synthesis of each stylized image proceeds through four sampling steps. We utilize the word "head" as the conditional text prompt, and specify the classifier-free guidance scale at 2. The style enhancement coefficient $\lambda$ is set at 1.2. These experimental procedures are conducted using a single NVIDIA 4090 GPU. Reference images are sourced from the AFHQ dataset [32], and content images are drawn from the CelebA-HQ dataset [21]. All images are processed at a resolution of 512 × 512 pixels.

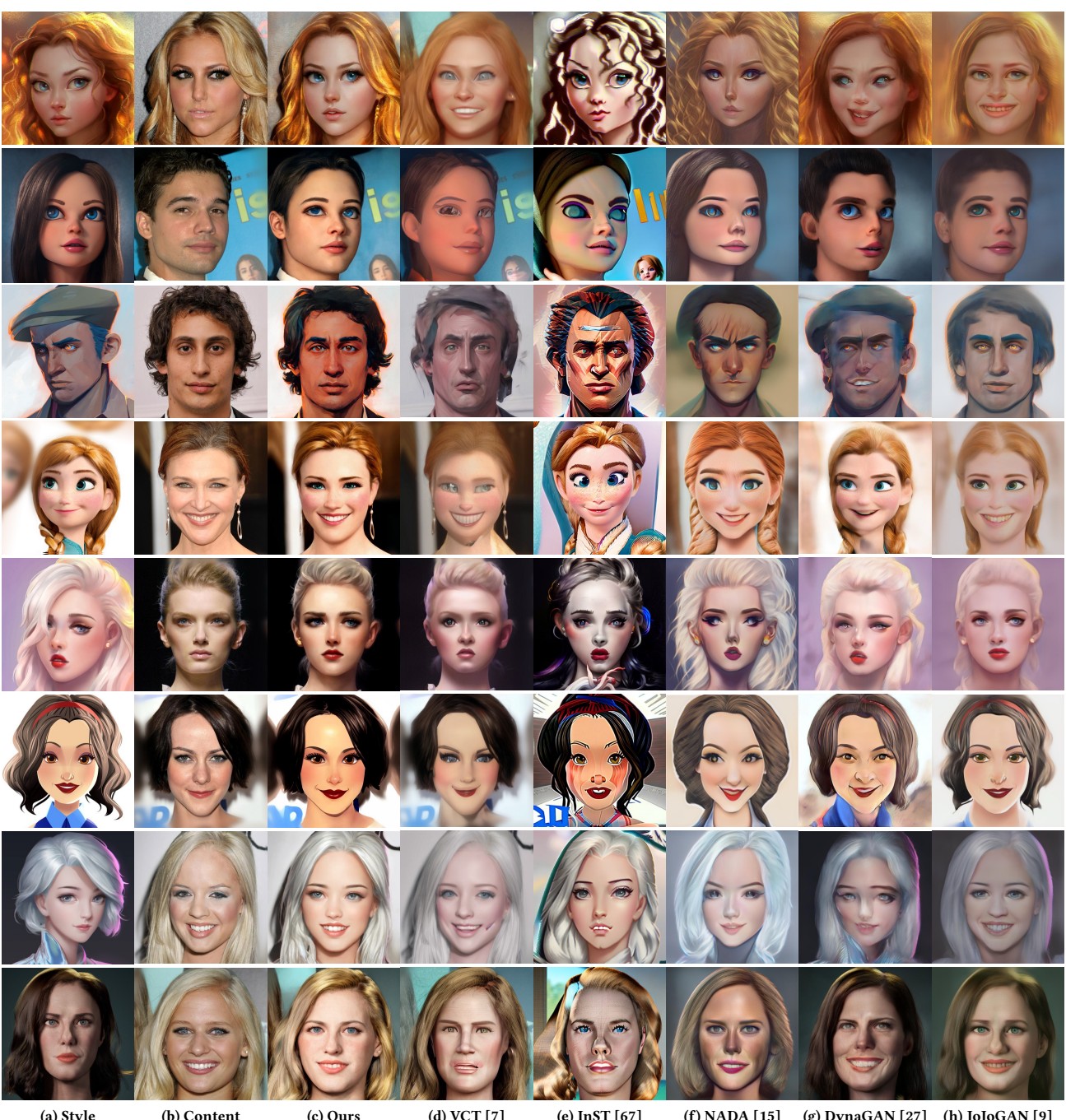

|  (a) Style | (b) Content | (c) Ours | (d) VCT [7] | (e) InST [67] | (f) NADA [15] | (g) DynaGAN [27] | (h) JoJoGAN [9] |

**Figure 4: Qualitative comparisons with conventional portrait stylization baselines. (a) and (b) are the input reference image and content image, respectively, while (c-h) are the stylized portraits synthesized by different baselines.**

## 5.1 Qualitative Comparison

**Baselines.** To evaluate our method's efficacy, we performed extensive comparative experiments against current state-of-the-art (SOTA) few-shot adaptation techniques. This analysis encompassed StyleGAN-based approaches, including JoJoGAN [9], StyleGAN

NADA (NADA) [15], and DynaGAN [27]. We also compared our method with diffusion-based techniques such as InST [67] and VCT [7]. All stylized outputs were generated using the respective open-source implementations provided by the authors.

**Table 2: Quantitative comparison with conventional portrait stylization baselines. The best and second best of each metrics will be highlighted in boldface and underline format, respectively. ↓ indicates the lower is better, and ↑ higher is better. The best and second best of each metrics will be highlighted in boldface and underline format, respectively. ↓ indicates the lower is better, and ↑ indicates higher is better. Our method achieved the best LPIPS and CLIP-CIA scores among all baselines, and the best Style score and fastest inference speed among all diffusion-based methods.**

|  | Methods | LPIPS ↓ | CLIP-CIA ↑ | Style ↓ | Fine-tuning(s) ↓ | Inference(s) ↓ |
|---|---|---|---|---|---|---|
| StyleGAN-Based | JoJoGAN [9] | 0.550 | 0.538 | 3.742 | 48.524 | 0.052 |
|  | DynaGAN [27] | 0.588 | 0.555 | **2.810** | 1156.822 | **0.041** |
|  | NADA [15] | 0.561 | 0.566 | 4.813 | 155.321 | 0.091 |
| Diffusion Based | InST [67] | 0.564 | 0.727 | 5.775 | 2007.966 | 6.932 |
|  | VCT [7] | 0.348 | 0.467 | 5.887 | 374.117 | 37.850 |
|  | **Our** | **0.261** | **0.858** | 5.213 | **0** | 0.684 |

Figure 4 provides a qualitative comparison among various methods. (a) presents reference artistic portraits, while (b) displays the original natural faces. And (c) illustrates the results of our ZePo, and the subsequent columns showcase outputs from various competing models. As illustrated in Figure 4 (f-h), although StyleGAN-based methods are effective in transferring the reference style to content images, they often lead to excessive stylization. This over-stylization results in significant deviations from the original content images, particularly altering facial poses as shown in the fourth and fifth rows for NADA [15]. Moreover, while JoJoGAN [9] achieves superior stylization effects compared to other StyleGAN-based methods, it struggles with content consistency, especially in preserving background elements of the content images. Among diffusion-based methods, InST [67] shows tendencies of overfitting, resulting in less desirable outputs. Conversely, VCT [7] manages a better balance between style transformation and content retention, though it often introduces significant changes in expressions. Our method not only facilitates various style transformations but also excels in preserving local details, such as facial features and hair texture, thereby maintaining consistent facial characteristics between the source and output images. For instance, specific local details, such as earrings in the first and fourth rows, are meticulously preserved in our methond.

## 5.2 Quantitative Comparison.

To demonstrate the superior quality and efficiency of our method in portrait art synthesis, we conducted quantitative comparisons with existing state-of-the-art (SOTA) methods.

**Metric.** To objectively assess the effectiveness of our proposed method, we employed LPIPS [66] for content preservation and VGG Style loss [16] for stylization evaluation. We observed that style loss predominantly focuses on external texture styles, which does not effectively capture the intrinsic style of images. Consequently, we propose the adoption of the non-referential evaluation metric, CLIP-IQA [56], for a more comprehensive assessment of image quality. CLIP-IQA leverages the CLIP model [44], pre-trained on a large-scale text-image paired dataset, as an image feature extractor. Then, this method evaluates the overall image quality through different text prompts that relate to image quality and aesthetics.

**Evaluation.** For quantitative assessment, we randomly selected 10 style images and 10 content images, generating a total of 100 stylized images for each baseline. The quantitative results are presented

in Table 2. Our method outperformed other techniques, achieving the best scores on both LPIPS and CLIP-IQA metrics. A lower LPIPS score indicates superior content preservation by our method, while a higher CLIP-IQA score reflects our method's ability to synthesize images with better overall quality and visual appeal. Additionally, our style score was the highest among methods based on diffusion models. In addition to evaluating the quality of the generated results, we assessed the fine-tuning and inference times required by each method, as presented in Table 2. The results indicate that the previous approaches necessitate extended periods for fine-tuning. Moreover, diffusion-based methods exhibit prolonged inference times, for example, InST [67] requires approximately 7 seconds to synthesize one stylized image, whereas VCT [7] experiences an increase in inference time to 37 seconds due to the need for Null-text text inversion [36]. Our framework, employing a zero-shot approach, eliminates the need for additional fine-tuning. By incorporating Style Enhancement Attention Control, we have reduced the inference time to approximately 0.6 seconds, thereby enhancing the practicality of our method.

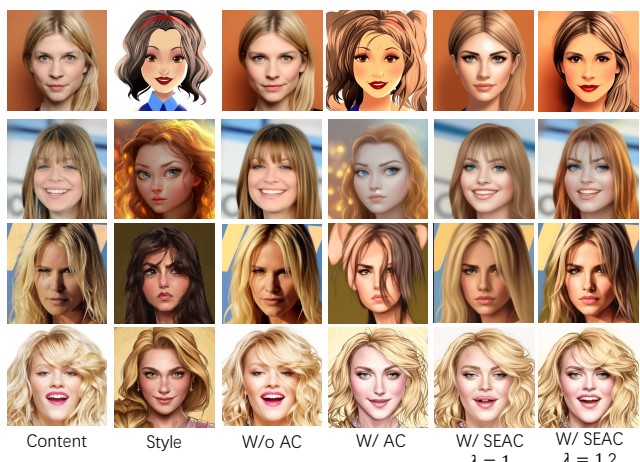

**Figure 5: Ablation experiments on Attention Control (AC).**

## 5.3 Ablation Study

**Attention Control.** We conducted extensive ablation experiments to verify the effectiveness of the proposed Style Enhancement Attention Control (SEAC). Figure 5 presents the ablation results using different Attention Control (AC) methods. Excluding AC results in merely the reconstruction of content images, lacking any substantive stylization. Conversely, the use of conventional AC often leads to over-stylization and the loss of critical content details. In contrast, our proposed Style Enhancement Attention Control (SEAC) maintains the integrity of content information while imparting a more subtle stylization effect. Additionally, the Style Enhancement (SE) coefficient effectively controls the strength of stylization. By adjusting the SE coefficient to 1.2, the stylization effect is notably enhanced, thus affirming the capability of SEAC to maintain a balance between content preservation and the desired level of stylization.

**Inference Steps** Figure 6 illustrates the results produced by our method at various sampling steps. Notably, our method can generate satisfactory stylized outcomes with just a single sampling step, and further increasing the number of sampling steps refines the detail of the synthesized images. As indicated in Table 3, enhancing the number of sampling steps leads to higher CLIP-CIA scores. However, this increment also results in a slight decline in content preservation and inference speed. To strike an optimal balance among stylization quality, content preservation, and inference efficiency, we established the sampling steps at four for all experiments. Additionally, we validated the effectiveness of the Feature Merge (FM) technique. As depicted in Table 3, implementing FM reduces the time required to synthesize images by 20%, without significantly compromising the quality of the generated images. This demonstrates that the feature merge technique not only enhances efficiency but also maintains high-quality stylization outcomes.

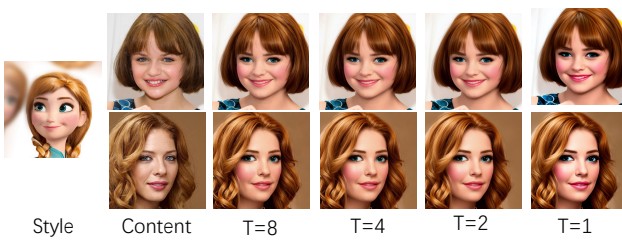

Style Content T=8 T=4 T=2 T=1

**Figure 6: Ablation experiment on different sampling time steps T. Our method can produce satisfactory stylized results with just one sampling, and further increasing the number of sampling steps can enhance the details of the synthesized results.**

**Consistency Features.** We performed an ablation study on the use of a fixed time step for consistent feature extraction in Figure 7. Contrary to initial expectations, extracting features directly from the input image without the addition of noise results in the inability of the model to discern content and style features effectively. This phenomenon is consistent with the behavior of the consistency equation (Eq. 4) at $t = 0$, where it merely outputs $z_0$ without any processing through the network. Consequently, during the consistency model distillation process, the noise prediction network $\epsilon_\theta$

**Table 3: Ablation experiment on different sampling time steps T and Feature Merge (FM). Increasing the sampling time steps can improve the CLIP-CIA score and reduce style loss. Employing FM can enhance the model's inference speed without significantly affecting the quality of the images.**

|        | Step  | CLIP-CIA ↑ | LPIPS ↓ | Style ↓ | Inference(s) ↓ |
|--------|-------|------------|---------|---------|----------------|
| W/ FM  | T=25  | 0.797      | 0.470   | **1.628** | 2.862        |
|        | T=8   | 0.835      | 0.409   | 1.848   | 1.049          |
|        | T=4   | 0.823      | 0.376   | 1.993   | 0.634          |
|        | T=2   | 0.781      | 0.326   | 2.181   | 0.416          |
|        | T=1   | 0.753      | 0.242   | 2.357   | **0.304**      |
| W/o FM | T=25  | 0.778      | 0.440   | 1.747   | 3.434          |
|        | T=8   | **0.844**  | 0.413   | 1.994   | 1.232          |
|        | T=4   | 0.817      | 0.368   | 2.094   | 0.724          |
|        | T=2   | 0.779      | 0.313   | 2.228   | 0.467          |
|        | T=1   | 0.766      | **0.226** | 2.411 | 0.355          |

has not learned to process inputs at $t = 0$ and therefore fails to extract features directly from $z_0$. Based on these ablation insights, we have set the fixed time step for consistency feature extraction to 99, enabling the extraction of more distinct consistency features.

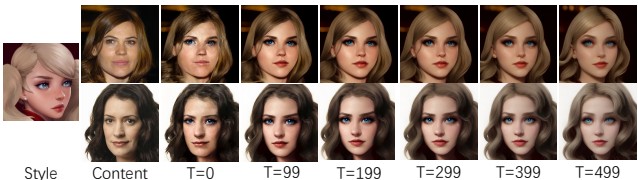

Style Content T=0 T=99 T=199 T=299 T=399 T=499

**Figure 7: Ablation experiment on the time-step $T$ during the forward noise addition process in consistency feature extraction.**

## 6 CONCLUSION

In this study, we introduce ZePo, a framework capable of rapidly generating stylized portraits. Unlike previous methods, ZePo eliminates the need for fine-tuning on specific samples or DDIM Inversion for input images, thereby enabling high-quality stylization in just four sampling steps. This efficiency reduces the inference time to merely 0.6 seconds. We also introduce Consistency Features extraction strategy, which leverage a pre-trained diffusion model to extract multi-scale Consistency Features from both content and reference images. Through the proposed style Enhancement Attention Control module, Consistency Features are adaptively fused, allowing for adjustable stylization intensity via the style enhancement coefficient. Furthermore, we introduced a feature merge technique to merge redundant consistency features, significantly decreasing the computational cost of attention control and enhancing the model's sampling speed. Extensive experiments demonstrate that our method can synthesize high-quality stylized results while effectively preserving the content integrity of the source image, markedly surpassing the performance of existing advanced approaches.

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
