# OpenReview forum: "ZePo: Zero-Shot Portrait Stylization with Faster Sampling"
_acmmm.org/ACMMM/2024/Conference — MM2024 Poster_

### Official Review · Reviewer_mJ9Z · 2024-05-05

**Rating:** 4
**Confidence:** 2

**Summary:**

This paper introduces ZePo, a portrait stylization framework that avoids DDIM inversion and relies on Latent Consistency Models (LCMs) to blend content and style features in just four sampling steps. Firstly, consistency features from both source and reference images are extracted using LCMs. These features are then utilized in the generation process of the final stylized portrait through the proposed style enhancement attention control module. Compared with current StyleGAN-based and diffusion-based methods, the proposed method shows better results, particularly in terms of LPIPS and CLIP-CIA scores.

**Strengths:**

1. The main methodology is easy to read and understand.
2. This method directly extracts features from the source and reference images using LCMs instead of common DDIM inversion, reducing time consumption.
3. Quantitative and qualitive results demonstrate that the proposed method is superior to other compared approaches.

**Limitations:**

1. It appears that CLIP-CIA is identical to CLIP-IQA in this manuscript; it is suggested to use one term consistently to avoid confusion. In addition, it is recommended to keep the names consistent such as x^{src} and I^{src} in Algorithm 1 and Figure 2. Moreover, there is a typo in Line 395: “is aims to” should be corrected.

2. The process of extracting consistency features at each transformer layer and the details of feature merging are not clearly explained. It would be beneficial to provide more detailed explanations for increased reproducibility. Specifically, clarifying whether the consistency features are the outputs of each self-attention layer would enhance understanding.

3. During the quantitative evaluation process, only 10 style images and 10 content images are utilized. Therefore, the numerical results in Table 2 are not very convincing.

**Suitability:**

2

---

### Official Review · Reviewer_S9bN · 2024-05-25

**Rating:** 4
**Confidence:** 4

**Summary:**

In this work, a zero-shot portrait stylization framework is proposed based on Consistency Models. The main component of this framework is a cross-image style feature injection module that utilizes self-attention features extracted from the Consistency Model during the inversion process as guidance. The primary contribution of this work is its leverage of the superior inference speed of Consistency Models compared to Diffusion Models, enabling more efficient portrait stylization.

**Strengths:**

As a test-time graphics problem, introducing Consistency Models in stylization for improved inference efficiency is a sound approach. Given that the output image’s style and content are determined by the reference and source images, the focus of this task should be on sample fidelity rather than mode coverage. Therefore, using Consistency Models is indeed a promising alternative to Diffusion Models. This approach offers valuable inspiration for future studies in image manipulation. I believe this work is near the acceptance bar for MM.

**Limitations:**

The technical contribution of this work is limited and incremental. The feature extraction and injection process using consistency models lacks an ablation study or insightful visualization to justify the current strategy. For instance, it is unclear in which part of the UNet (e.g., encoder layers, decoder layers, and different resolutions) the self-attention features contribute the most to stylization. Additionally, there is no analysis of redundant features from the self-attention layers used in the current setup. Why should Self-Attention K and V features be used for style injection instead of others? Further justification on this is needed.

Moreover, the Self-Attention feature injection strategy closely follows Prompt-to-Prompt (P2P) [3]. The use of self-attention K and V feature injection is the same as in [1]. The feature merging strategy has been employed by other works, such as RIVAL [2]. The style enhancement module mirrors the reweighting strategy used in P2P. The authors should modify the motivation extraction to place more emphasis on the Consistency Model itself, which is well-justified. Additional discussions on the technical setbacks encountered when adapting attention-based control strategies from Diffusion Models to Consistency Models would be beneficial.

[1]Alaluf, Yuval, et al. "Cross-image attention for zero-shot appearance transfer." arXiv preprint arXiv:2311.03335 (2023).

[2]Zhang, Yuechen, et al. "Real-world image variation by aligning diffusion inversion chain." Advances in Neural Information Processing Systems 36 (2024).

[3]Hertz, Amir, et al. "Prompt-to-prompt image editing with cross attention control." arXiv preprint arXiv:2208.01626 (2022).

**Suitability:**

2

---

### Official Review · Reviewer_RL3T · 2024-05-25

**Rating:** 4
**Confidence:** 2

**Summary:**

The paper proposed ZePo, a framework for zero-shot portrait stylization that leverages diffusion models to synthesize stylized facial images without requiring model fine-tuning. ZePo achieves efficient content and style feature fusion in just four sampling steps, utilizing Latent Consistency Models and a novel Style Enhancement Attention Control mechanism to speed up the process. Extensive experiments demonstrate that ZePo significantly enhances the efficiency and fidelity of portrait stylization, achieving high-quality results with minimal inference time.

**Strengths:**

This paper is well-organized and straightforward. The experiments clearly demonstrate the advantages of the proposed pipeline. Leveraging the LCM improves the inference time significantly.

**Limitations:**

[1]. How does $\lambda < 1$​ impact the visual results?

[2]. Is the SEAC also available for text embedding guidance?

**Suitability:**

2

---

### Official Review · Reviewer_HWco · 2024-05-26

**Rating:** 3
**Confidence:** 3

**Summary:**

This paper presents an inversion-free portrait stylization framework based on diffusion models that accomplishes content and style feature fusion in merely four sampling steps.

**Strengths:**

1.Proposes an inversion-free portrait stylization framework, enhancing image generation speed.
2.Achieves content and style feature fusion in just four sampling steps.
3.Employs Latent Consistency Models with consistency distillation to effectively extract Consistency Features from noisy images.
4.Introduces a Style Enhancement Attention Control technique for meticulous content and style feature fusion.

**Limitations:**

1.Unlike traditional style transfer methods that use VGG for feature extraction, the authors utilized VAE to obtain Consistency Features in section 4.1. I would like to understand the advantages of using VAE for the style transfer task or the reasons for not using models like VGG here. It is well-known that VAE sacrifices high-frequency features; therefore, I hope there are relevant experiments to validate this approach.

2.Is the Enhancement coefficient learnable? Is there any relevant experiment to prove the reliability of Enhancement coefficient?Why can SEAC effectively assess the similarity?

3.Can the models compared in Table 2 give more annotations? I don't know if this is the result of several sampling steps? According to Table
3, different number of steps does not seem to have a direct effect.

**Suitability:**

2

---

### Meta-Review · Area_Chair_75mZ · 2024-06-27

**Recommendation:** Accept (Poster)
**Confidence:** 4

**Metareview:**

After the rebuttal, the reviewers unanimously recommend the acceptance of this paper. Therefore, we are happy to recommend the acceptance of this paper.